# Transport of the Hunga volcanic aerosols inferred from Himawari-8/9 limb measurements

Fred Prata [1,2]

[1]AIRES Pty Ltd, Mount Eliza, Victoria, Australia. Correspondence: Fred Prata (fred@aires.space)
[2]School of Electrical Engineering, Computing and Mathematical Sciences, Curtin University, Kent St, Bentley, Perth, WA 6102, Australia

**Abstract.** The Hunga volcano (21.545 ºS, 178.393 ºE; also known as Hunga Tonga-Hunga Ha'apai) erupted on 15 January 2022 producing copious amounts of aerosols that reached high into the stratosphere, exceeding 30 km and settling into layers a few kilometres deep between 22–28 km. The Advanced Himawari Imager (AHI) on board the geostationary Himawari-8/9 platform at 140.7 ºE was able to monitor the eruption at 10 minute intervals and 0.25 $km^2$ to 4 $km^2$ spatial resolution within 16 spectral channels ranging from visible to infrared wavelengths and over a latitude/longitude field of view of ~±75º. Here a new use of these data is proposed where the limb region of the field of view is exploited to detect aerosol layers extending vertically into the atmosphere. The analyses provide vertical profiles of scattered visible light and are compared to Caliop space lidar measurements. Hunga aerosols are detected using the ratio of near infrared reflectances at 1.61 $\mu$m and 2.25 $\mu$m, in the western limb from 22 January and in the eastern limb from 31 January 2022 up until the present time ( December 2023). The average zonal velocity is estimated to be ~–25 $ms^{-1}$ (westwards) and the meridional velocity to be ~0.2 $ms^{-1}$ (northwards). The latitudinal spread is characterised by a gradual northerly movement of the main layer situated between 22–28 km in the first 60 days, and stagnation or slight southerly spread thereafter. There is a shallow maximum of the lower stratospheric aerosol between 10–20 °S and the aerosol loading during 2023 is elevated compared with the 3 months prior to the eruption. The southern hemisphere (0–30 °S) tropical lower stratospheric aerosol e-folding time is estimated to be ~12 months, but the decay is not uniform and has high variability. The current methodology does not provide quantitative estimates of the amount or type of aerosol, but based on the spectral properties of water and ice clouds the analysis suggests there is a strong liquid water content in the aerosol layers.

## 1 Introduction

The Hunga eruption (Van Eaton et al. (2023), also known as Hunga Tonga-Hunga Ha'apai eruption) was the most energetic eruption in over a century and possibly more energetic than the major eruption of Krakatau in August 1883 (Wright et al., 2022; Matoza et al., 2022). The main explosive event occurred at around 04:15 UTC on 15 January 2022 and sent material (volcanic ash, water and gas) high into the atmosphere, reaching heights >30 km with some material reaching as a high as 57 km into the mesosphere (Proud et al., 2022; Carr et al., 2022). The explosion also generated acoustic-gravity waves and a surface propagating wave–the Lamb wave–that travelled across the globe several times (Wright et al., 2022; Otsuka, 2022;

Purkis et al., 2023; Vergoz et al., 2022). An unprecedented amount of water vapour (Millan et al., 2022), was deposited into the stratosphere from the eruption, increasing the climatological mean amount by up to 50% and forecast to persist for several years (Millan et al., 2022). Other gases, mostly in the form of $SO_2$ were also emitted (Carn et al., 2022) and rapidly converted to sulfate acid aerosol (Sellitto et al., 2022; Zhu et al., 2022).

The transport of the Hunga aerosol has been modelled (Legras et al., 2022) observed from the ground (Baron et al., 2022; Boichu et al., 2023) and observed by satellite instruments, the Caliop lidar (Sellitto et al., 2022; Boichu et al., 2023), the Infrared Atmospheric Sounder Instrument (IASI) (Sellitto et al., 2023) and the Ozone Mapping and Profiler Suite (OMPS) (Taha et al., 2022), among others. The water vapour plume has been measured by the Microwave Limb Sounder (MLS) (Schoeberl et al., 2022). Geostationary satellite instruments, such as the Meteosat Second Generation Spin Enhanced Visible and Infrared Imager, the Geostationary Operational Environmental Satellite Advanced Baseline Imager and the Himawari-8/9[1] Advanced Himawari Imager (AHI) include many narrowband channels (visible and infrared) that can be used to observe and measure volcanic aerosols directly below as they pass under the instruments' field of view. Such measurements include visible aerosol optical depth and various qualitative products, such as the RGB (Red-Green-Blue) ash/$SO_2$ product (see https://user. eumetsat.int/resources/user-guides/ash-rgb-quick-guide.) In this work, a method is presented to detect and monitor aerosols that extend several kilometres high into the atmosphere, by exploiting measurements of the scattered light from the limb view of the AHI on board the Himawari-8/9 geostationary.

## 2 Methodology

The eruption and its effects were measured by several Earth orbiting satellites, including the AHI (Gupta et al., 2022) that has a continuous view of a part of the Earth extending ~75 °W, °E, °S and °N of its sub-satellite location. The AHI instrument (see https://www.data.jma.go.jp/mscweb/en/himawari89/space_segment/spsg_ahi.html) is stationed ~35,795 km over the equator at 140.7 °E, and consists of 16 narrowband channels measuring reflected and emitted light (0.4–13.6 $\mu m$) from the Earth and atmosphere at spatial resolutions ranging from ~0.25 $km^2$ (~0.64 $\mu m$), 1 $km^2$ (~0.47, 0.51 and 0.86 $\mu m$) to ~4 $km^2$ (~1.61–13.3 $\mu m$ in 12 separate bands) at the nadir point. Data are acquired over the Earth disk at 10 minute intervals. The AHI is designed so that a small portion of the field of view contains pixels extending beyond the Earth's limb into space (see Fig. 1). Occasionally this allows a view of solar system objects, such as the Moon (Nishiyama et al., 2022) (as can be seen in Fig. 1), and stars (Taniguchi et al., 2022) to be imaged by the AHI's 16 spectral channels. Horváth et al. (2021), and Proud (2015) and Tsuda et al. (2018) have used the limb view of geostationary satellites to determine the height of volcanic clouds and mesospheric clouds, respectively, at high latitude locations in the northern hemisphere. The work of Horváth et al. (2021) is particularly relevant here as the limb viewing geometry for a geostationary satellite is provided in detail there. For the purpose of this study, the equations for determining distances and locations of AHI views are based on Horváth et al. (2021) and were coded using the Python 3 programming language. A full mathematical description of the viewing geometry can be found in Horváth et al. (2021), and here for completeness only the most relevant equation is included.

---

[1]In future Himawari-8/9 will be referred to as Himawari.

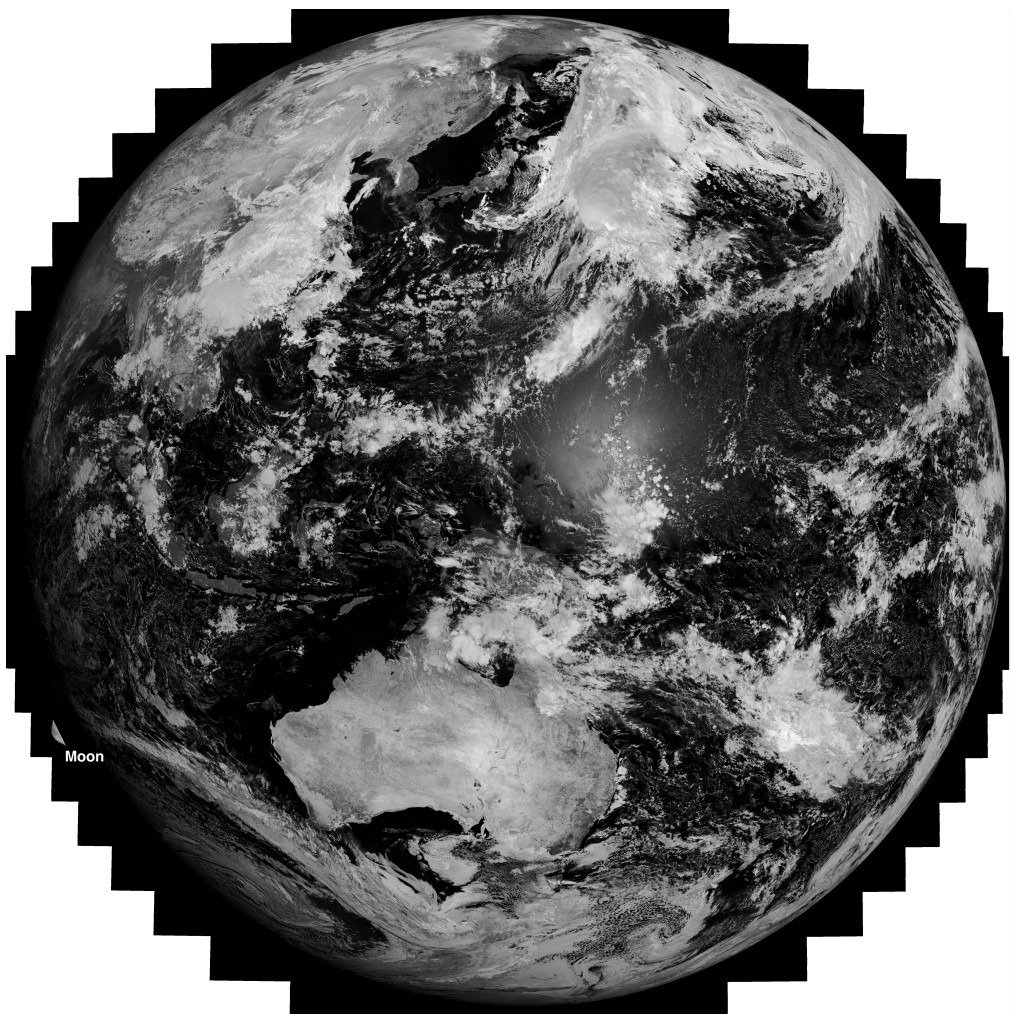

**Figure 1.** Field of regard of the Himawari Advanced Himawari Imager (AHI) shown for a single spectral channel (band 6). The black coloured region around the Earth's limb is also imaged. The feature seen at bottom-left is part of the Moon. Himawari AHI data courtesy JMA/JAXA.

## 2.1 Limb geometry

The AHI scans the Earth's disk every 10 minutes starting in the north. The total angular field of view in the E-W and N-S direction is 17.6º resulting in a single pixel instantaneous field of view of 13.97 $\mu$rads for a total scan of 22000 x 22000 pixels, corresponding to the resolution of the AHI visible channels (see Japan Meteorological Society (2017)). The scan geometry is such that a small portion extends beyond the Earth's limb and out into space. For the purpose of this study the direction of the limb is defined as all light paths originating from space that reach the satellite instrument without intersecting the Earth.


A simple calculation, using the scan geometry information, is used to determine the distance of these 'space pixels' from the Earth's limb and after correcting to the local vertical an estimate of the height above the Earth's limb can be found. The relevant equation is,

$$h = |S_B| \arccos\left(\frac{S_B \cdot S_P}{|S_P||S_B|}\right),$$  (2.1)

where $|S_B|$ is the distance (corresponding to the look vector, $S_B$) from AHI to a point on the Earth's limb, and $|S_P|$ is the distance to a point passing through the unit vector extending from $P$ outwards, and intersecting another point on the limb (see Fig. 2). Horváth et al. (2021) shows (see Appenidix A) that $|S_P|$ can be calculated from the known scan angles, assuming an Earth ellipsoid (GRS80), and $|S_B|$ is calculated from the pixel geodetic latitude and longitude provided with the image data. A correction is applied to convert this tangent height to a local vertical height ($h$), as explained by Horváth et al. (2021) this is a small correction of a few 100 m's. All of the processing of the image data was done using the Python programming language

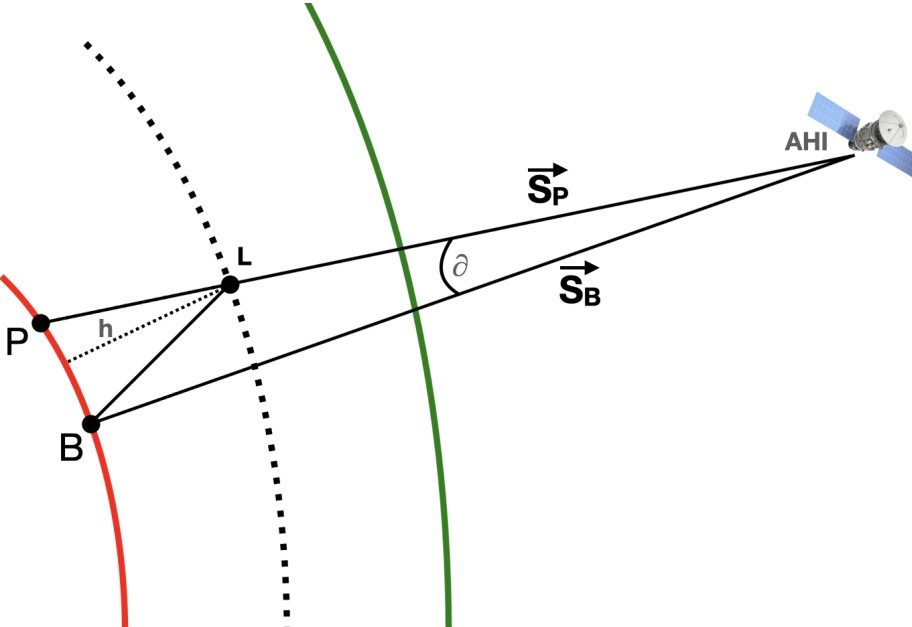

**Figure 2.** Illustration of the limb geometry calculation. The look vectors $S_B$ and $S_P$ enclose the angle $\partial$ which can be used to calculate the distance from P to L. Here $h$ is the local vertical height and the dashed-black line indicates an observation plane on which L lies, within the atmosphere. The red and green lines represent the surface of the Earth and the edge of the atmosphere, respectively.

using publicly available packages, particularly satpy (Raspaud et al., 2018). The AHI data already contains the necessary information for locating the Earth's limb as the data are supplied with a complete grid of latitudes and longitudes, which are assigned the value of NaN (Not a Number, according to the satellite data provider) for pixels not on the Earth's surface. This affords a simple scheme for locating the locus of limb pixels which involves processing the image location data line-by-line across the scan to find the first occurrence of a NaN. The edge of the atmosphere is located by extending outwards by ~86

pixels which was found to correspond to pixels with 0.45 $\mu$m reflectance values <1%. Since the radiometric uncertainty for this channel is ~3% (see http://www.data.jma.go.jp/mscweb/en/himawari89/space_segment/doc/AHI8_performance_test_en.

pdf).), reflectances<1% should guarantee that the path does not include any atmosphere or surface. A slice of the image along the limb (the '*limb image*') is extracted for each of the visible and near-infrared channels at 0.47, 0.51, 0.64, 0.86, 1.61 and 2.25 $\mu$m and one infrared channel at 11.2 $\mu$m (that is, the highest spatial resolution visible and near-infrared channels and one infrared channel), and then rotated (for convenience in viewing) and replotted onto a latitude-height coordinate system. The longitude of the point corresponding to the vector from the spacecraft to the line extending from the Earth limb point is not

necessarily the same as the longitude of the aerosol, since the aerosol is optically thin (at the wavelengths considered) and could be extensive in the E-W (or W-E) direction. Looking eastwards the longitudes are east of ~144 °W, and looking westwards they are west of ~66 °E, and vary by ~3° in longitude from 0–30 °S latitude. No correction was made for refraction of the visible light from the limb image as this error is both difficult to correct and is smaller than potential geolocation errors due to operational AHI image navigation (see Horváth et al. (2021) for more detail).

## 2.2  Generating the limb image

The processing of the AHI data to obtain the limb image is illustrated in Fig. 3, containing an RGB true-colour rendition of the raw data near the limb (Fig. 3(a)), an extracted portion, rotated with the locations of the limb and estimated extent of the atmosphere indicated (Fig. 3(b)), and finally the limb image in the latitude-height coordinate system using a ratio reflectance measure (Fig. 3(c)) – see next section. It should be noted that the estimated extent of the atmosphere above the limb is an

arbitrary choice; it was determined using a threshold value on reflectance and to ensure that at it was at least a distance of 40 km above the Earth's limb. The radiative transfer of light rays from the limb to AHI of this unusual use of AHI data, as a vertical limb sounder, has not been fully explored here and instead only the utility of using the data for delineating aerosol layers and investigating the temporal character are investigated. One possible way to investigate the transport of scattered light from the Earth's limb to the Himawari satellite, is to use a limb scattering radiative transfer code, such as SASKTRAN Bourassa et al.

(2008). A goal then, is to find a measure that best discriminates aerosol layers from background light in the limb view.

## 2.3  Spectral discrimination

In the interests of brevity, it was decided to investigate the information content of the data first and postpone radiative transfer modelling to a later study. AHI has several channels that are sensitive to aerosol and cloud properties. Generally, AHI data have been used to study aerosols and clouds using data obtained from Earth observing geometry, which is how the imaging system

was designed to operate. Many studies have been developed to infer aerosol and cloud properties supplemented with radiative transfer modelling based on this viewing geometry (Yoshida et al., 2018; Su et al., 2021). These studies suggest that, against a dark background (e.g. the ocean surface), aerosol layers cause an increase in scattered light measured at the satellite. Against the dark background of space a similar increase in scattered light should occur. After experimenting with various combinations

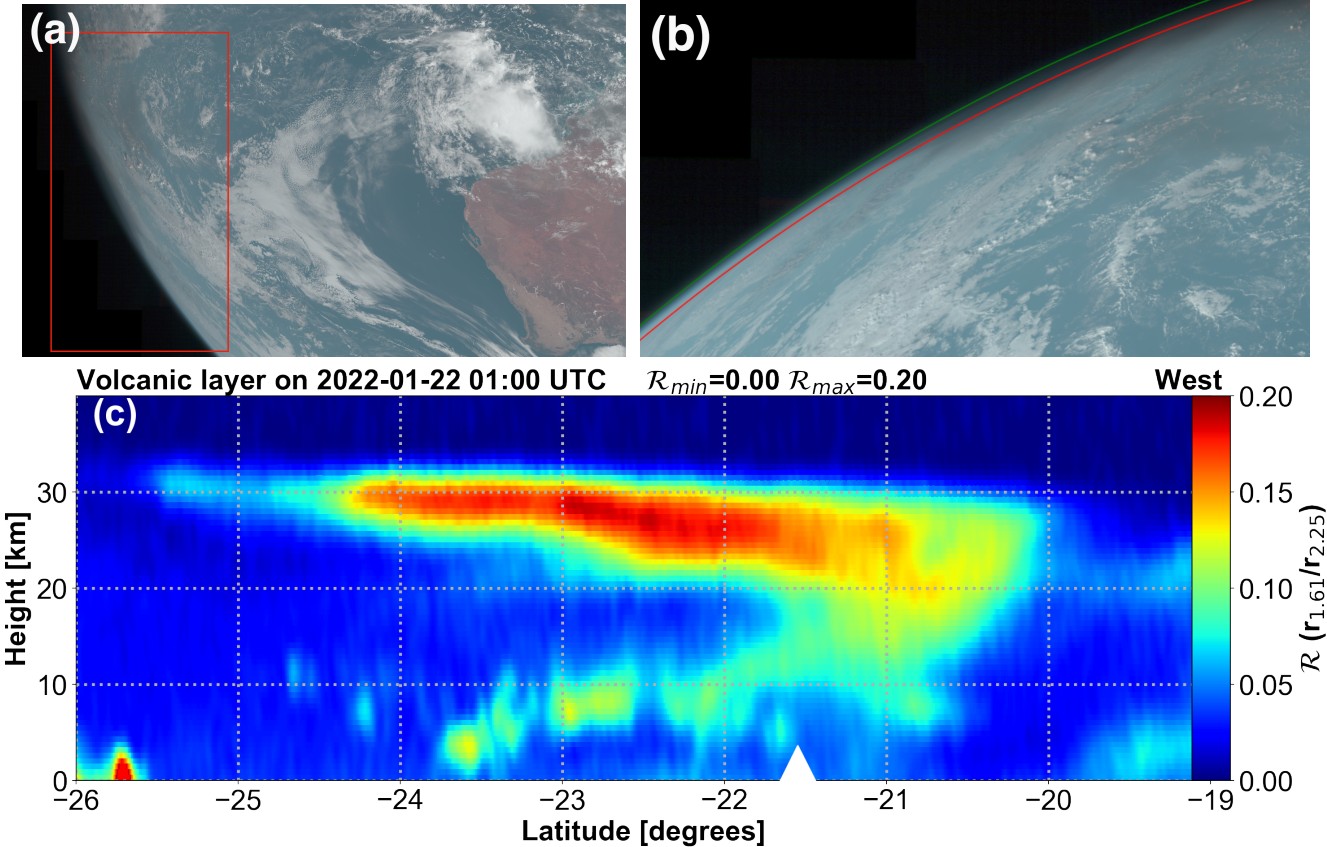

**Figure 3.** (a) *Top*: True-colour (RGB) AHI image on 22 January 01:00 UTC. (b) *Middle*: Rotated image corresponding to the region indicated by the red-coloured rectangle. The red and green coloured lines show the limb and extent of the atmosphere, respectively. (c) *Bottom*: Limb image (Latitude-height) using a ratio of channels (1.61 $\mu$m / 2.25 $\mu$m) to delineate stratospheric aerosols, looking towards the west (longitudes ~62–65 °E). The white triangle shown is at the latitude of the Hunga volcano. Himawari AHI data courtesy JMA/JAXA.

of the AHI visible and near infrared channels the ratio, $\mathcal{R}$ was found to be a good discriminator of Hunga aerosols, where:

$$\mathcal{R} = \frac{r_{1.61}}{r_{2.25} + r_0},$$
(2.2)

and $r_{1.61}$ is the reflectance[2] (%) in AHI channel 5, centred at 1.61 $\mu$m and $r_{2.25}$ is the reflectance at 2.25 $\mu$m (channel 6). The constant $r_0$ is a positive (>0) arbitrary reflectance at 2.25 $\mu$m used to ensure that the ratio is well defined (to avoid divide by 0). The logic behind the choice of this ratio is as follows. For deep space, both $r_{1.61}$ and $r_{2.25}$ tend to small values (<1%), with $r_{1.61}$ usually reaching its noise value first. Thus for clear skies $\mathcal{R} \to 0$. When aerosols are present, the ratio is > 0 and its

magnitude can be used to distinguish between predominantly liquid or ice aerosol content. This is based on previous published studies using this ratio (Miller et al., 2014; Zhou et al., 2022; Noh et al., 2019). Specifically, if the aerosol is predominantly

---

[2]The term *reflectance* is used rather than scattered light only to refer to the data and not to the light transport process

liquid water then $\mathcal{R}$ tends to be larger than if the cloud were predominantly ice. A value of $r_0 = 10\%$ was found to ensure a clear sky ratio of <0.01. The discrimination of ice from water using the 1.61 and 2.25 $\mu$m channels relies on differences in the imaginary parts of the refractive indices of ice and water between these two channels. To illustrate this feature, Fig. 4 shows

the ratio of the imaginary parts of the refractive indices of ice (Warren and Brandt, 2008) to water (Segelstein, 1981) plotted as a function of wavelength from 1–2.5 $\mu$m. The normalised spectral response functions for the two AHI channels used are also shown on the plot. It can be seen that the ratio is much higher at 1.61 $\mu$m compared to at 2.25 $\mu$m. The higher the ratio is, the greater the absorption due to ice. Thus larger values of the reflectance ratio, $\mathcal{R}$, suggest lower refractive index ratio and hence indicate less ice and more liquid water. This analysis was repeated using refractive index data (Palmer and Williams, 1975)

(obtained from the Oxford University ARIA database: https://https://eodg.atm.ox.ac.uk/ARIA/; last access: 4 April 2024) for sulfate aerosol and the ratio constructed with respect to the imaginary part of the sulfate refractive index (dashed line in Fig. 4). The same conclusion can be drawn that a higher vale of $\mathcal{R}$ suggests more liquid water (or sulfate aerosol) than ice. To

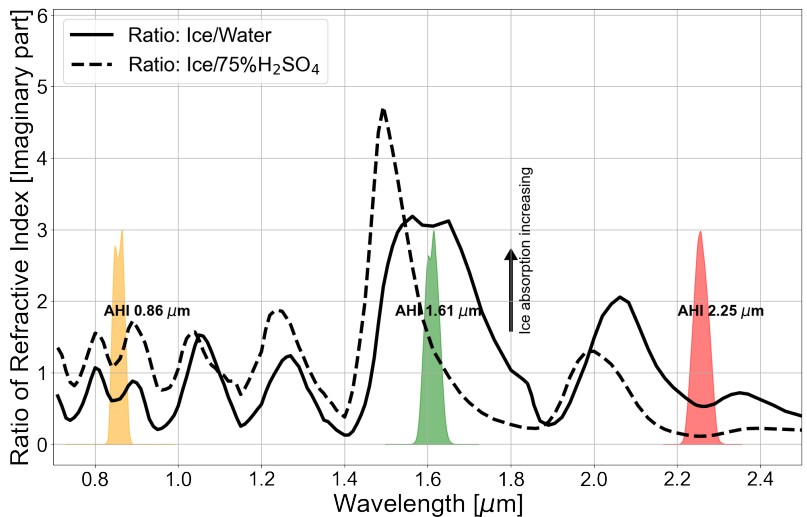

**Figure 4.** Ratio of the imaginary part of the refractive index of ice to water (solid line) and ice to sulfate (dashed line). The locations of the normalised spectral response functions of AHI channels at 0.86 (orange), 1.61 (green) and 2.25 $\mu$m (red) are also shown.

make this analysis more quantitative requires radiative transfer modelling, which will be the subject of future work. Here it is argued that this simple ratio contains information on aerosol type. Observations of the Hunga aerosol from balloon-borne *in*

*situ* measurements suggest that the aerosol particles are small (radius < 1.0 $\mu$m), transparent and slightly absorbing, suggesting sulfates and liquid water (Kloss et al., 2022). Duchamp et al. (2023) used SAGE III (Stratospheric Aerosol and Gas Experiment) to find sulfate particles with mean effective radius of 0.4 $\mu$m. The large amounts of water vapour from the eruption that entered the stratosphere have been estimated to be up to ~150 Tg (Millan et al., 2022; Zhu et al., 2022) and so the Hunga stratospheric aerosol is likely to have a high water content.

## 2.4 Detection geometry

The movement of the volcanic aerosol layer combined with its vertical extent suggest that it may not always be detected using limb viewing, and because visible reflectance data are used, the time of measurement is also important. At the start of the eruption the aerosol was observed to travel westwards in a plume-like manner. The rate of travel was approximately ~18 degrees longitude per day (~25 ms$^{-1}$) (Legras et al., 2022), thus after 6–7 days the aerosol layer should be detected looking towards the western limb, while no aerosol layer should be seen looking eastwards. Fig. 3(c) shows the aerosol layer on 22 January at 01:00 UTC looking towards the western limb and Fig. 5 shows no aerosol layer on 21 January at 21:00 UTC (4 hours earlier), looking eastwards. After two weeks the aerosol layer will have circulated the globe at 20 ºS and hence be

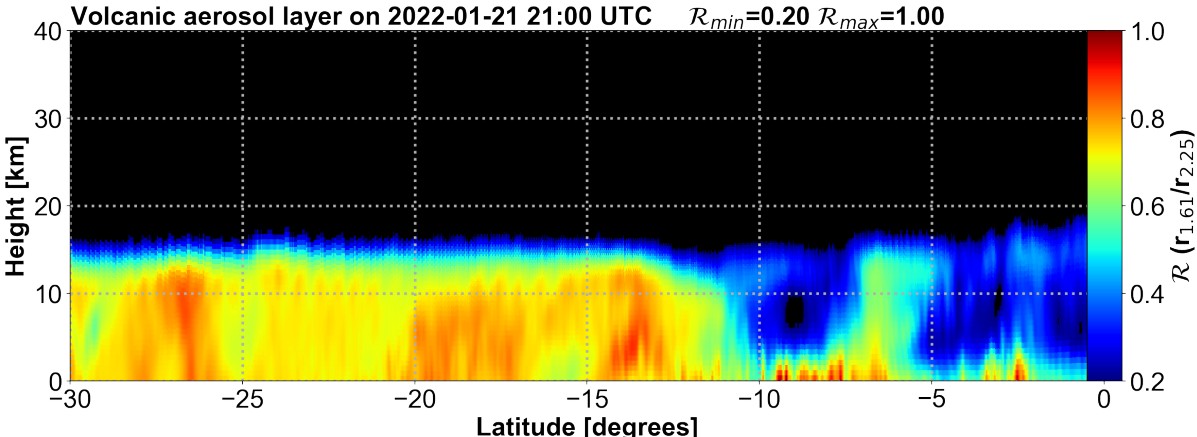

**Figure 5.** Limb image (Latitude-Height) for 21 January 2022 at 21:00 UTC looking towards the eastern limb. As expected, no aerosol is detected in this image.

detectable in views towards the east and west.

## 3 Results

Data were processed for each day looking toward the eastern limb at 21:00 UTC (to ensure enough sunlight was available) from 10 January to 30 April 2022 and then every 5 days of each month from April 2022 until December 2023. These data were sufficiently sampled in both temporal resolution and duration to determine the main characteristics of the spread of the aerosol.

### 3.1 Vertical information

Vertical cross-sections through the aerosol layer were calculated as a function of latitude each day (at 21:00 UTC) using the reflectance ratio metric ($\mathcal{R}$). Four consecutive days in February 2022 are shown in Fig. 6. The Himawari limb images can be compared with satellite lidar measurements of stratospheric aerosols by the Caliop instrument in polar orbit (Winker et al.,

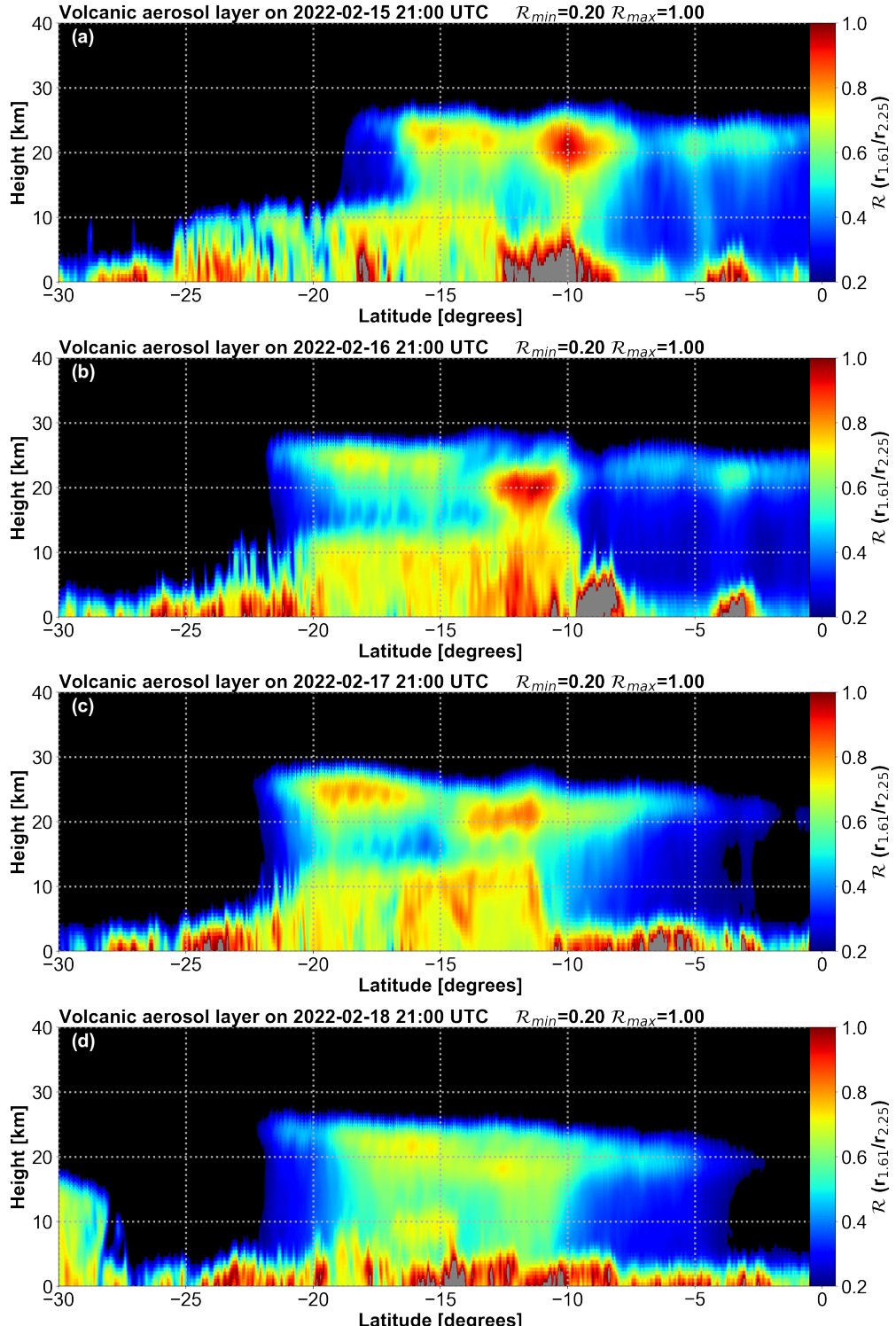

**Figure 6.** (a)–(d) Vertical height cross-sections through the aerosol layer as a function of latitude for 15–18 February 2022. Grey coloured areas are where $\mathcal{R} > \mathcal{R}_{max}$. All views are looking towards the eastern limb direction. $\mathcal{R}$ is the ratio of reflectances at 1.61 $\mu$m and 2.25 $\mu$m.

2009). Caliop makes vertical soundings of backscattered light at two wavelengths and two polarisations which can be used to distinguish aerosol types, including volcanic aerosols (Prata et al., 2017). Detailed comparisons between the limb data and Caliop measurements are complicated by the different geometries which make spatial and temporal coincidence problematic.

Qualitative comparisons suggest a good degree of agreement as shown in Fig. 7 where a Caliop 'curtain' on 7 February is plotted above a Himawari limb vertical cross section. The highest layer is found at around 25–28 km between ~15–25 ºS, and a lower, thinner layer at around 22 km between ~2–15 ºS. The limb image, while coarser in vertical resolution and noisier, also detects these upper (~25 km) and lower (~22 km; 8 °S) layers at similar latitudes. There are also several differences between the Caliop and Himawari cross-sections: notably the layers are deeper in the limb data and multiple thin layers in

the Caliop measurements near 20 °S are absent in the Himawari measurements. Mishra et al. (2022) also show examples of Caliop detections with thin layers ~<2 km deep. These differences are likely due to the quite different viewing geometries, with Caliop's nadir viewing geometry and the Himawari limb view observing the layers at oblique angles. The different viewing geometries make intercomparisons between measurements from the two sensors difficult to interpret in a quantitative manner and have not been pursued.

## 3.2 Time series

The decay of the stratospheric aerosol to background, climatological conditions is expected to take several years (Millan et al., 2022). Observations of the water vapour amount in the Southern Hemisphere at 26 hPa (~25 km) from the Microwave Limb Sounder show a significant increase from climatology from the start of 2022 up to the end of 2023 (Rozanov et al., 2024; Nedoluha et al., 2024) (see also: https://acd-ext.gsfc.nasa.gov/Data_services/met/qbo/qbo.html). To investigate the decay of

the aerosol, the reflectance ratio used here was averaged from ~18–28 km and from 30 °S to 0 °S and plotted as a function time. The Himawari data were sampled at 21:00 UT at 5 day intervals (1 day intervals from 10 January 2022 to 30 April 2022) and the mean (open circles) and $\pm 2\sigma$ (shaded light-grey) of $\overline{\mathcal{R}}$ (averaged over latitude and height) are shown in Fig. 8. The rapid increase in the ratio $\overline{\mathcal{R}}$ can be seen ~7 days after the Hunga volcano erupted, followed by high variability into the middle of 2022. Up until mid-August 2022 there appears to be a quite rapid decrease in $\overline{\mathcal{R}}$ and the vacillation seen in the time-series

is presumably due to zonal transport (see next subsection), as higher concentrations of the aerosol pass out of the limb view of Himawari. Prior to January 2022, the $\overline{\mathcal{R}}$ background level is ~0.04 and up until the end of 2023 the values have not quite reached the background level. The e-folding time for an exponential decay with $k^{-1}$=12 months is shown on the plot (solid black line), with the green and red lines show $k^{-1}$= 6 and 24 months, respectively. The e-folding time is within the range usually assumed for a tropical eruption injecting aerosols into the stratosphere (Kremser et al., 2016). However it is possible

that changes in the composition of the aerosol are affecting the sensitivity of $\mathcal{R}$, and also meridional transport may be affecting the calculation, which only considers latitudes from 10 °–30 °S. The meridional variation of $\overline{\mathcal{R}}$ (averaged over time) from 0–30 °S in the 3 months prior to the eruption, for 2022 and for 2023 is shown in Fig. 9 where the mean and standard deviation of $\mathcal{R}$ over the three time periods was calculated in 2 degree latitude intervals from ~18–28 km. Assuming $\overline{\mathcal{R}}$ is proportional to the aerosol optical depth, the aerosol has a broad peak between 18–20 °S during 2022 and has spread to the equator and 30 °S, in

agreement with other results, see for example Schoeberl et al. (2023). During 2023 the aerosol is still evident, as $\overline{\mathcal{R}}$ is elevated

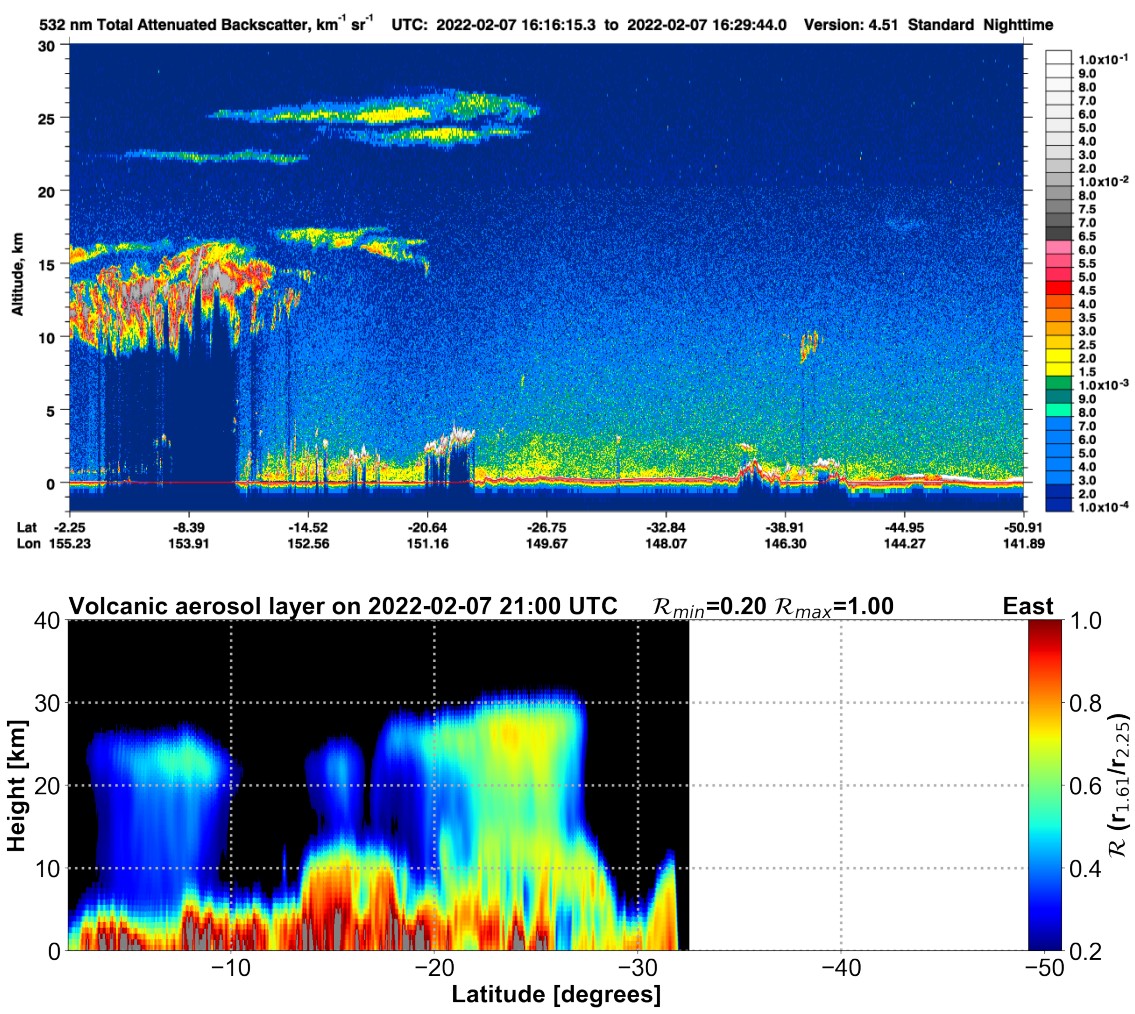

**Figure 7.** *Top panel:* Caliop 'curtain' showing 532 nm backscattered light on 7 February 2022 between 16:16–16:30 UTC. *Lower panel:* Himawari limb image on 7 February at 21:00 UTC, looking towards the east. Note that the latitude scale on the Himawari limb image has been reversed (decreasing to the right) and the range increased to match the Caliop curtain.

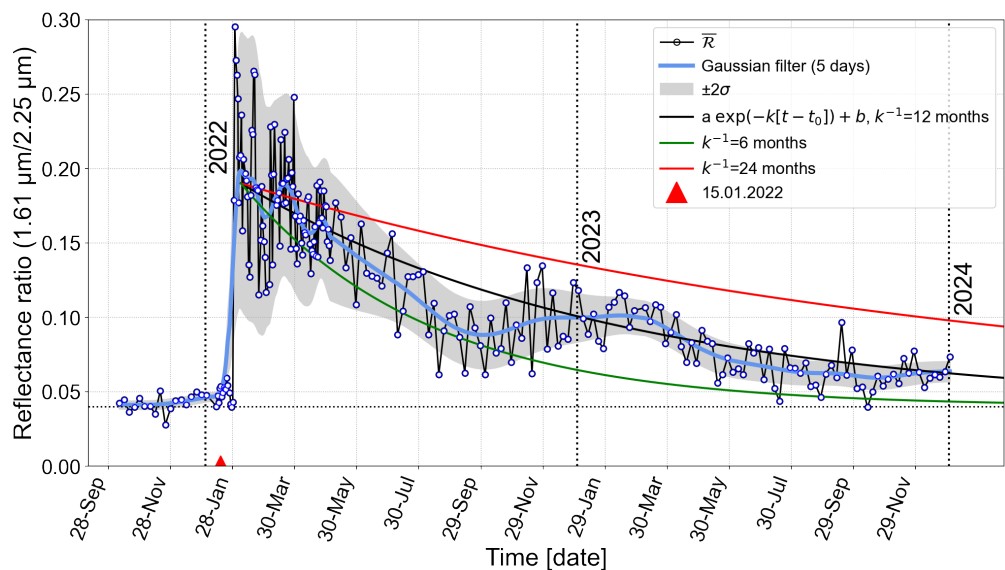

**Figure 8.** Time-series of $\overline{\mathcal{R}}$ from 1 October 2021 until 31 December 2023. The temporal sampling is 5 days, except from 10 January until 30 April 2022 when it is 1 day. The data are averaged over the height range ∼18–28 km and latitude range of 0–30 °S. The shaded light-grey region encloses ±2$\sigma$ of the smoothed (Gaussian filter with a filter width of 5 days) series (light blue line). Also shown is an exponential decay curve (black line) with a time constant, $k^{-1}$ = 12 months, and green and red lines showing $k^{-1}$ = 6 and 24 months, respectively. The parameters used are: $b$=0.04 (the background value of $\overline{\mathcal{R}}$), $a$=0.15 and $t_0$=30 January 2022. The start date of the Hunga volcanic eruption is indicated by a red triangle.

compared to its value prior to the start of the eruption, but is now more evenly spread. The slight rise in $\overline{\mathcal{R}}$ at higher southern latitudes in the 3 months prior to the eruption and during 2023, are apparently not related to the Hunga volcanic eruption.

### 3.3 Time-Height-Latitude spread

The latitudinal spread of the aerosol was studied by averaging the $\mathcal{R}$ height profiles into four layers: 14.3–18.5, 19.0–22.1,
22.7–26.8, 30.3–33.9 km for each day at 21:00 UTC from 15 January to 15 September, 2023. The latitudinal range considered was from the equator to 30 °S. The layers were selected to represent the region above most clouds to the tropical tropopause, the lower and mid-regions of the lower stratosphere and the upper stratosphere. The results are summarised in Fig. 10 as time-latitude plots of $\overline{\mathcal{R}}$, each shown with the same range of values of $\overline{\mathcal{R}}$ from 0 to 1. The results indicate the strongest signals in the regions above the tropopause and in the lower stratosphere, with almost no signal in the upper stratosphere (heights > 30
km) ∼3 weeks after the eruption. The general spatial patterns agree well with those reported by Legras et al. (2022), Nedoluha et al. (2024) and Rozanov et al. (2024) using OMPS and Caliop observations, which show the aerosol mainly confined between ∼22–28 km and 30°S to 0°. The meridional spread of the aerosol is very slow compared to the zonal spread. The peaks in $\overline{\mathcal{R}}$ are interpreted as the locations of densest aerosol and appear at each latitude every ∼ 16 days or so, as they circumnavigate the

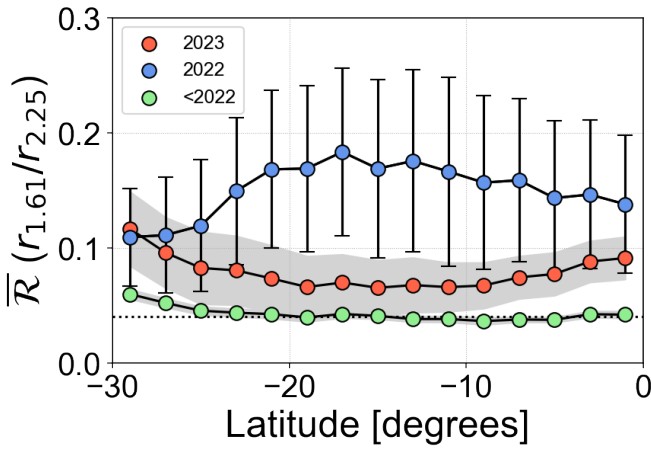

**Figure 9.** The meridional variation of $\overline{\mathcal{R}}$ for three time periods: October–December 2021 (green circles), January–December 2022 (blue circles), and January–December 2023 (red circles). The light-grey coloured regions enclose $\pm 1\sigma$ of the means, and the vertical (error bars) show $\pm 1\sigma$ of $\overline{\mathcal{R}}$. The dotted line indicates the assumed background level of $\overline{\mathcal{R}}$ (=0.04).

Earth. Because of the viewing geometry the data sampling is such that the aerosol is observed appearing over the eastern limb of the Earth and then continuing to move westwards; a peak in the detection occurs each time the main layers appear in the eastern limb. Presumably some aerosol is still present but the signal strength (as measured by $\mathcal{R}$) is not sufficient to identify it. Under these assumptions it is possible to estimate the zonal ($u$) and meridional ($v$) velocities of the maximum aerosol concentration in the lower-mid stratosphere. The zonal velocity is estimated as $u \approx 23\text{–}28 \text{ ms}^{-1}$ which is reasonably consistent with zonal wind speeds as these latitudes and heights during January-April. There is coherence between the spatial patterns of the aerosol in the first three height layers, perhaps suggesting that the aerosol is quite vertically thick. There is also an indication of some vertical separation with parts of the layer moving more rapidly zonally, than other parts. The meridional velocity $v \approx 0.2 \text{ m s}^{-1}$, which is in good agreement with monthly mean values determined from ERA-5 re-analysis data (Hersbach et al., 2023) as shown in Fig. 11. The upper stratospheric easterly jet slowly descends from 5 hPa in January to about 30 hPa by April and also migrates northwards. This has the effect of retarding the westward movement of the aerosol at ∼20 km and more southern latitudes. The meridional spread is strongest northwards between January–February but the spread of the aerosol's progress equatorwards is slowed thereafter as the winds are close to zero or turning negative (southwards). These general observations qualitatively agree with the spread observed in the Himawari limb imagery (see Fig. 10); however, both $u$ and $v$ are poorly constrained by the measurements and the quantitative values should only be regarded as approximations.

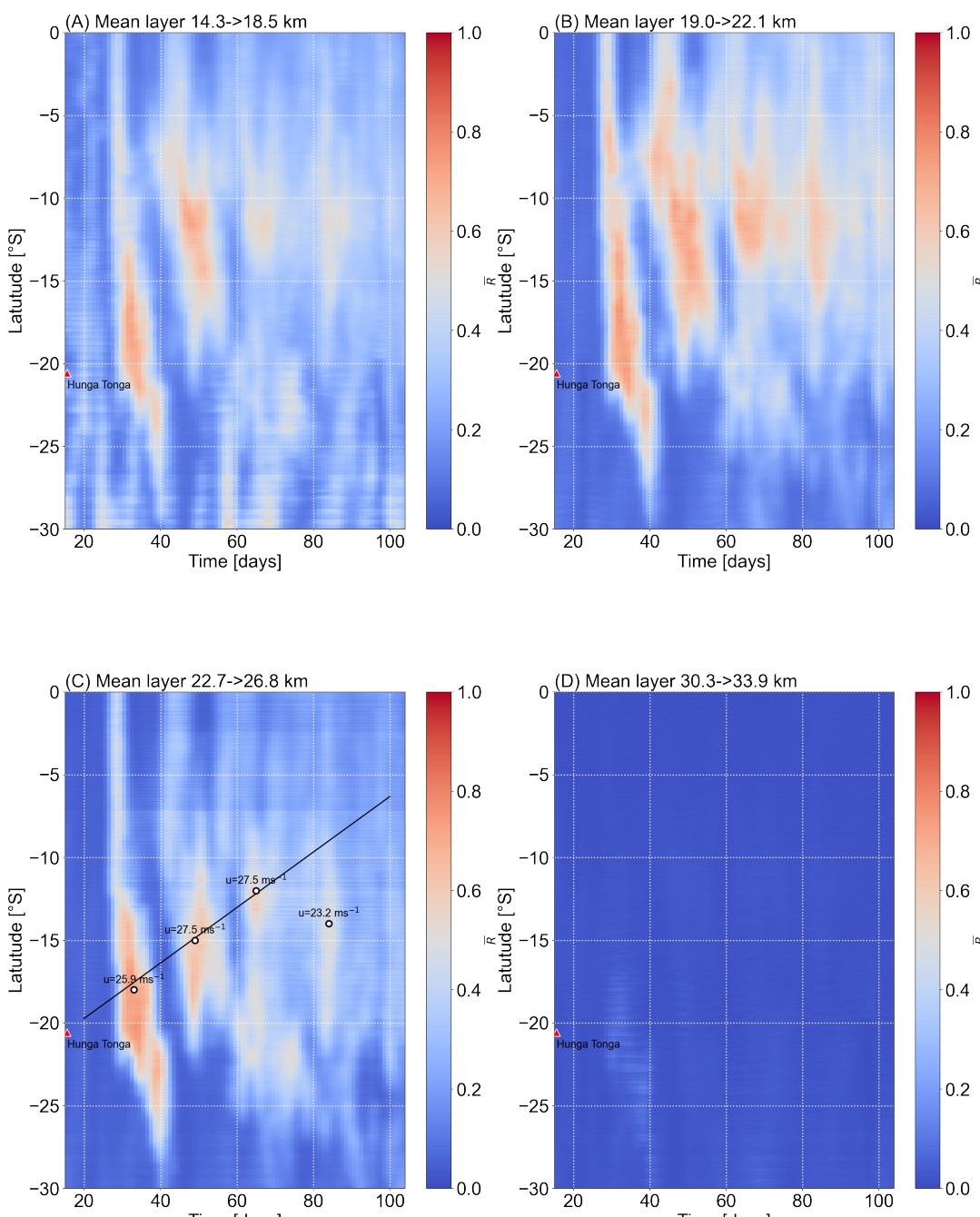

**Figure 10.** Time-Latitude plots of $\overline{\mathcal{R}}$ for four different height ranges: **(A)** below the tropopause, **(B)** in the lower stratosphere, **(C)** in the mid-stratosphere, and **(D)** in the upper stratosphere.

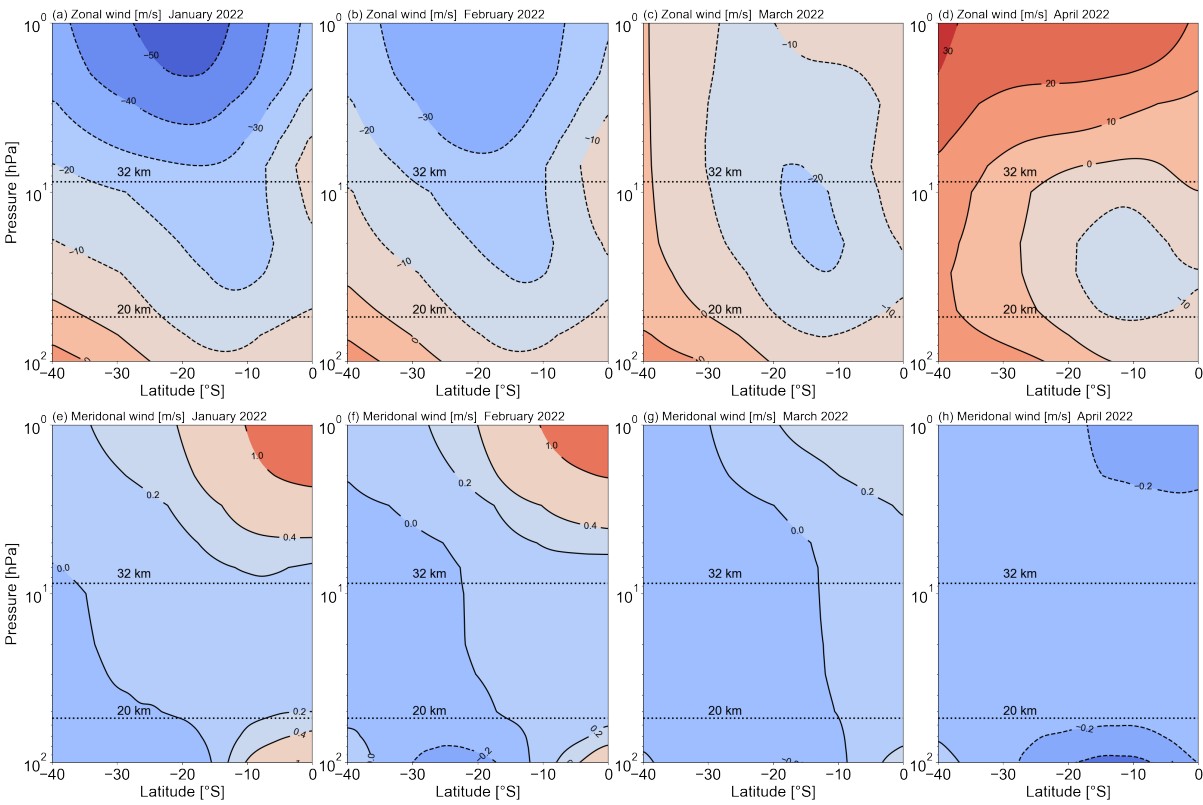

**Figure 11.** ERA-5 re-analysis monthly mean wind ($u$, zonal: upper panels (a)–(d); $v$, meridional: lower panels (e)–(h)) wind fields as a function of pressure (15 levels) and latitude (2.5 ° resolution) for January, February, March and April, 2022.

## 4   Conclusions

The eruption of the Hunga volcano on 15 January 2022 is likely to have been the most energetic volcanic eruption since Krakatau in 1883 and certainly since the start of the satellite era in the ~1960s (Matoza et al., 2022; Proud et al., 2022; Wright et al., 2022). The eruption sent up to ~150 Tg of water vapour (Millan et al., 2022) and ~0.4–0.5 Tg of $SO_2$ (Carn et al., 2022), resulting in 0.66±0.1 Tg of sulfuric acid (Duchamp et al., 2023) sent into the stratosphere where it formed layers ~2–6 km thick over a height range of ~22–28 km. The aerosol was first observed in Himawari limb measurements on 22 January 2022 and

could still be detected in December 2023, 23 months later. The measurements presented here suggest that during the first three months, the aerosol travelled zonally at speeds of ~-25 ms$^{-1}$ (westwards) and meridionally at <0.5 ms$^{-1}$ (northwards) in broad agreement with ERA-5 re-analysis winds for the same time period, latitudes and heights. Very little transport into the northern hemisphere was detected with the aerosol mainly confined between latitudes 0–30 °S. These results are in good agreement with Mishra et al. (2022) using SAGE-III/ISS, Calipso data and a Lagrangian transport model. The limb measurements have quite

limited vertical resolution but there are some indications of multiple layers, which can be clearly seen in Caliop data.

The ratio of reflectance at 1.61 $\mu$m and 2.25 $\mu$m, used to identify the aerosol, suggests that the aerosol contains liquid water rather than ice but no further investigation of the aerosol composition has been made. An assumed exponential decay of the stratospheric aerosol, as represented by the mean reflectance ratio $\overline{\mathcal{R}}$, has an e-folding time of ~12 months. Exploitation of other AHI channels might provide greater insight into the aerosol properties but this will require some rigorous radiative transfer modelling which is beyond the scope of this paper.

*Data availability.* CALIOP data are available through the NASA Langley Research Center Atmospheric Science Data Center, https://asdc.larc.nasa.gov/ (last access: 21 October 2023). Himawari data are available from amazon web services: https://noaa-himawari8.s3.amazonaws.com/index.html (last access: 31 October 2023). ERA-5 re-analysis data are available from: https://cds.climate.copernicus.eu/cdsapp#!/dataset/reanalysis-era5-pressure-levels-monthly-means?tab=form (last access: 22 September 2023).

*Author contributions.* FP wrote the paper, devised the methodology, wrote the code and analysed the data.

*Competing interests.* The author declares no competing interests.

*Acknowledgements.* The author is most grateful to Chris Boone for a careful review of the manuscript and for some insightful comments. I am also grateful to the constructive and helpful comments from two anonymous reviewers and to Luis Millan Valle who corrected a reference to the MLS data . Their time and suggestions are appreciated. The Himawari-8/9 data were provided by the Japanese Meteorological Agency and the Japanese Aerospace Exploitation Agency. Dr Andrew Prata is thanked for useful discussions and for assistance with accessing and processing the satellite data. The author acknowledges use of the ERA-5 data contains modified Copernicus Atmosphere Monitoring Service information (2023) from the Copernicus Climate Change and Atmosphere Monitoring Services. The author also thanks the developers of the Python satpy package and for making it freely available to the research community.

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
