# Peer review of "Transport of the Hunga volcanic aerosols inferred from Himawari-8/9 limb measurements"

_EGUsphere, 2023_

## Author Response (AR1)

Responses 1.

Dear Editor, dear Authors,

The manuscript "Transport of the Hunga Tonga volcanic aerosols inferred from Himawari-8 limb measurements" applies existing methodologies to derive qualitative information on the stratospheric aerosol load from limb observations at the peripheric field of view of Himawari observations to Hunga Tonga-Hunga Ha'apai (HTHH) eruption in January 2022. The topic of the manuscript is of interest for the AMT readership. The study itself is quite short. The main result, besides the fact that the HTHH eruption signature is visible from Himawari limb observations, is an estimation of the plume speed as it spreads zonally and meridionally. I would consider the possibility to add the words "plume speed" or similar in the title. The paper is relatively clear, even if some aspects should be clarified (see Specific Comments). More efforts are needed to compare Himawari observations with simultaneous CALIOP observations of the HTHH plume, and plume speed results with what is already present in the literature (again, see Specific Comments). Besides these points, I don't actually have major concerns, and, in my opinion, this manuscript should be published when the following Specific Comments are addressed.

My best regards

Thank you for the comments and the time you have taken to consider my paper.

I think the use of "plume" for the aerosol is arguable and the word "transport" conveys a better description of the westward movement and N-S spreading. I have removed the word "Tonga" from the title (for reasons explained in my response below). The Specific Comments are very helpful and I have accepted nearly all of them. My responses below.

Specific Comments:

1) L23: "persisting up to the present time (September 2023)" can you please add a reference for that? As far as I know, there are no long-term observations published on water vapour perturbation from HTHH eruption.

I've added a link to a NASA site that shows those observations and referred to some recent papers on this. Actually, this paper also provides those observations. I have added the text to reflect that it is predicted the aerosol will persist for some years – reference included. I have also added two new Figures which illustrate the spread of the aerosol through time and space up until the end of 2023.

2) L24: "...and converted... please add a statement on the rapid nature of this conversion (Zhu et al., 2022 or Sellitto et al., 2022 already cited in your manuscript)

Added the word "rapidly" and reference as suggested.

3) L25: "observed from the ground..." what about AERONET analyses of Boichu et al. (https://agupubs.onlinelibrary.wiley.com/doi/10.1029/2023JD039010?af=R)? For the satellite dispersion you might mention the IASI (already shown by Sellitto et al. 2022 and Legras et al., 2022 but also more systematically here: https://essopenarchive.org/users/527694/articles/656444-observing-the-so2-and-sulphate-aerosol-plumes-from-the-2022-hunga-tonga-hunga-ha-apai-eruption-with-iasi)

Thanks.  I missed that reference but have now added it in.

4) L29: the phrasing "limb viewing *aspect*" should be clarified already at this point (even if this is clearer later in the text, i.e. in the methodology section)

I have clarified what is meant by "limb viewing" and placed this as a new sentence in the section "Limb geometry" – which I think is the best place.

5) L45-46: how much is the resolution in km?

AHI resolutions are now included in a new paragraph under "Methodology".  The resolutions vary depending on spectral channel.

6) L53-54: the sentence is not fully clear to me

The detail was deliberately excluded from this paper as it is elegantly explained by Horvath et al. (2021).  The correction arises from the fact that the view is not strictly at 90 degrees so there is a very slight foreshortening in h.  I feel this is well explained by Horvath and the point of mentioning this here was to alert the reader to the full detail in Horvath's paper.  However, I have added further explanatory sentences as I agree this part needed more information.

7) L56-58: "...which are assigned...limb...of a NaN." is it too technical for this kind of manuscript? This can probably be deleted.

I have retained this but defined the meaning of NaN.  I like to keep this as detection of these values is part of the algorithm.

8) L59-60: this is also difficult to understand for me. Can you please clarify?

I have added an explanation in the revision. By setting a lower bound on the reflectance it is hoped that clear space is being detected with no atmosphere – so this provides an upper height value.

9) L60-on and Eq. 2.2: even if previous works are cited here adressing this issue, more details should be added to the reasons for the choice of the two wavelengths in the definition of the parameter R. Is this choice supported by RTM simulations? Why not other channels? E.g. the IR channel at 11.2 microns is located in a sensitive band for H2SO4 (see Sellitto and Legras 2016 or Sellitto et al. 2017), can any information on the HTHH plume be extracted using this channel?

Agreed. I have added an explanation and a new Figure that supports the use of the R ratio. I did look at one infrared channel (11.2 μm) but essentially this does not give better information than the ratio. The problem is that this is an emission measurement (with cold space) behind the aerosol so the sensitivity does not appear to be good enough.

10) Figure 2: not clear what red and green curves are. Is it the same as Fig. 3b? Please explain and add this information in the caption.

Agreed. I have added definitions in the caption.

11) Figure 2 caption: delta angle symbol is not consistent in the figure and caption

Thanks. Fixed.

12) L67: "in a sequence of three panels (Fig. 3)" --> "in Fig. 3"

Thanks. Changed.

13) L88-89: by the way, how can R be negative?

Yes my sentence is misleading. I have re-worded this to read >0 and degree of "positiveness".

14) L92-93: please have a look at this for satellite observations of the particle size distribution of HTHH plume:
https://agupubs.onlinelibrary.wiley.com/doi/full/10.1029/2023GL105076

Thanks. Reference added in two places (here and in Conclusions).

15) L118-119: please also discuss the many differences between Himawari and CALIOP observations in Fig 6

Agreed. I have added sentences describing the main differences and speculated on the reasons – mostly due to differences in viewing geometries. I have also added another reference.

16) Is there anything more that can be done in terms of comparison of limb Himawari and CALIOP observations? Please consider more systematic comparisons, which is basically the only "major" comment that I have for this manuscript.

Yes I am sure there is. Unfortunately, I think this would require a lot of work and a new paper. The problem is trying to find coincidences and accounting for the different viewing geometries. It would be a major work but is do-able and would also require some radiative transfer calculations to make the R-ratio more quantitative.

17) L121-122: why this choice of altitude intervals? How they represent what's mentioned at L123-124? What if different choices of altitudes ranges are chosen?

I did try different intervals but found only small differences. The ranges were informed by analysing numerous vertical profile plots at various latitude locations and also to look at different layers within the stratosphere.

18) L137-138: this is discussed in Legras et al., 2022; please use their work to discuss your results: are the vertical spread and different velocities pointed out by Legras et al., 2022 consistent with what you get?

Yes. The Legras et al. paper is referred to just above (L127) where it is mentioned that these results agree with Legras. So yes they are consistent.

19) Fig. 5 and others: can also a longitude information be added to this kind of figures - like done for CALIOP curtains e.g.?

Not really. The limb view covers a range of longitudes but I have included the two limits and stated the range in variation of longitude (somewhat similar to how Caliop samples the Earth below).

20) Fig 7: why not putting letters to identify panels (panels a, b, c and d)?

Yes. I have added letters (A), (B) etc as suggested.

21) Fig 7: it is somewhat surprising that the aerosol signal in Himawari limb observations shows up only starting from about 1 month after the eruption. A number of previous works have shown that the formation of secondary sulphate aerosols was very quick for this eruption. How can you explain this? Can you, e.g. a very quick apparation of large signal in the lower stratosphere in the western limb,

as expected considering the HTHH literature and, in particular, CALIOP, OMPS and IASI observations of Legras et al., 2022 and Sellitto et al., 2022?

The first detection is on 22 January (just 1 week after the eruption)-see Fig. 3. But the aerosol layer is moving, taking ~16 days to circumnavigate the globe (at 20-30 S). The limb detection has some lower sensitivity (which I have not estimated), so even if there is aerosol at all longitudes along these southern latitudes, it is only detected if sufficiently optically thick.

22) L147: "largest" considering which metrics?

Well there are several metrics and it's probably the largest according to many of them. I have changed "largest" to "most energetic" – and I think that is a defendable statement (I've added a reference).

23) Throughout the whole paper and also in the title: please use the full name of the volcano "Hunga Tonga-Hunga Ha'apai" or, after defining it, the abbreviation "HTHH".

Not agreed. As explained by Van Eaton et al (2023), the islands Hunga Tonga and Hunga Ha'apai are part of a much larger (submerged) caldera belonging to the Hunga volcano. So I have changed "Hunga Tonga" to "Hunga", added the Van Eaton et al (2023) reference and included "Hunga Tonga-Hunga Ha'apai" in two places as this seems now to have become common usage.

Responses 2.

This paper presents an interesting study on the ability of Himawari-8 limb observations to provide information on the vertical profile of aerosols injected into the atmosphere by volcanic eruptions, such as that of Hunga Tonga in early 2022. The results are convincing about the possibility to use geostationary satellite measurements for such purpose.

Thank you for the comments and your time to consider my paper.

However, I have the following reservations before publications in Atmospheric Measurement Techniques:

- Other aerosol products based on Himawari-8 measurements are available, e.g. the RGB-Ash product (https://navigator.eumetsat.int/product/EO:EUM:DAT:MSG:VOLCANO/print) that has been used to analyze Hunga Tonga aerosols. Such products should be mentioned in the introduction and contrasted to the aerosol product presented in this study, which focuses on the aerosol vertical distribution.

  I have added a sentence about aerosol products from geostationary satellite instruments.  I don't think the RGB ash product is particularly relevant for this study.

- The paper is loosely written and often lacks details that would help better understand some aspects of the study. For example, the description of the Advanced Himawari Imager (AHI) instrument is lacking and there is few information on the measured radiances and their characteristics. Only the limb geometry is explained in section 2, which relies largely on a study by Horvath et al. (2021) without a summary of the corresponding methodology. As an example, the parameter "h" in equation 2.1 is not precisely defined and it is not clear also what is considered to be the Earth limb. In satellite limb measurements the limb corresponds to the whole atmosphere viewed by the instrument, which does not seem to be the case here. A more thorough definition of the various parameters and notions used in the study is thus needed.

  Yes it was an oversight not to include some discussion of AHI in the main text (it is in the Abstract).  I have rectified that by adding two sentences (in Methodology) and a reference to the Himawari Users Guide.

  I have also added more text, 3 new Figures which I hope explain the analyses and results more clearly.

I have defined h and the meaning of "limb". Horvath et al. (2021) provide a very detailed analysis of the geometry of limb viewing using geostationary instruments and I do not see a need to repeat that in this paper.

- Some characteristics of the study are given without explanation. For example, in page 3, line 59, it is not clear why the edge of the atmosphere corresponds to pixels with 0.45 μm values <1%. Similarly, why only 7 of the spectral channels are used and what is behind the choice of the mentioned channels?

  Yes. I have added an explanation of the use of <1% and added a reference. Only 7 channels were considered as the other 9 channels are either redundant or not useful. I have added a sentence to provide a reason for this selection.

- Even if the longitude of the points considered in the analysis is undefined a possible range of longitudes could be indicated by taking into account the viewing geometry of the satellite instrument.

  Yes. I have included the eastern and western longitude limits as suggested and added a range for the eastern and western look directions.

- More information is needed on the limits of the methodology described in the paper for the determination of aerosol vertical profile. In which altitude range is it the most efficient? What is the accuracy of the R product introduced in Equation 2.2? More information is also required on the choice of the reflectance at 1.61 and 2.25 μm. Some references are mentioned but overall the explanations are not sufficient.

  I have added a Figure (new Figure 4) which explains the principle for using channels at 1.61 and 2.25 μm and explanatory sentences. It's not clear to me what is being asked about the efficiency of the altitude range? Essentially the altitude resolution is determined by the inherent pixel size, the angular sampling and the viewing geometry. It is difficult to assess the accuracy of the R product without having something to compare it against. Caliop is a good candidate for this but the differences in viewing geometry between Caliop and Himawari make this difficult to interpret in a quantitative manner. I have added sentences which explain this. I guess some careful radiative transfer calculations would help determine an accuracy for R but I feel this is beyond the scope of this short paper, which was simply to introduce the use of limb measurements from an unlikely source.

- Since the aerosol cloud takes some time to form from the $SO_2$ cloud, despite more rapid formation linked to increased water vapor amounts due to the

Hunga Tonga eruption, results of the study should be analyzed in this context. How is the method sensitive to the initial evolution of the aerosols from initial ash to ice and liquid aerosols? How is the estimated meridional velocity in section 3.2 affected by possible artefacts in the method?

Actually, it seems that as the sulfate aerosol formed quite rapidly, detecting it after 7 days seems quite plausible.  I don't think there was much ash in the dispersing cloud, certainly nothing like that expected from such a large eruption.  Therefore I don't think it is necessary to discuss these processes in any detail.  It is known (from the recent literature) that the aerosol formed quickly and consisted of predominantly sulfate/liquid water aerosols.  The main sensitivity of the method is to optical opacity of the aerosol.  As explained, initially (less than 1 week) no aerosol is seen in the either limb and this is likely because it was not dense enough to detect and also it had not had enough time to reach the western or eastern limbs.  The first detection on 22 January is towards the west; a few hours before looking east there is no aerosol.  Thereafter, as the aerosol travels around the Earth, almost periodic peaks in the aerosol are observed (every 16 days or so). Now there may well be other aerosols (smoke from fires, other eruptions, convective outflows into the lower stratosphere) that contribute to changes in the stratospheric optical depth.  The ratio was chosen as it is sensitive to liquid water and ice and so is mostly detecting the Hunga Tonga aerosol. The averaging process, time period, coherence and agreement with ERA-5 climatology (and other work, e.g. Legras et al) give confidence that any artefacts due to the methodology are minimal.

I have added two new Figures: one showing the temporal evolution of the aerosol (as measured by the reflectance ratio) and one showing the meridional spread over the 26 months of data analyses.  I think these Figures help to demonstrate that the methodology has some value in assessing the transport and spread of the Hunga aerosol.

**Minor comments**

- The figures' legends are generally incomplete. For instance, legend of Figure 1 does not describe the red, green and dashed black lines. Legend of Figure 5 does not mention the limb (East or West).

Thanks.  Descriptions of the red, green and dashed black lines are included in the Figure caption.  The direction (

- The initial letter of "Earth" should be capitalized throughout the paper.

Thanks.  Fixed.

---

## Referee Report (RR1)

This article describes observations of the Hunga Tonga volcanic plume using high resolution imager measurements from a geostationary satellite. In general, this is a clever approach for extracting atmospheric aerosol information, a bonus set of information that was presumably not targeted in the original mission design. However, there are a few aspects of the interpretation that I don't entirely agree with.

The signals are referred to as reflectances. I expect this comes from standard usage of the data products for Earth observation. The measured signals actually come from scattering by the atmospheric aerosols, rather than reflection, along the lines of limb scattering missions such as OSIRIS. The abstract briefly mentions "scattered light," but for me the terminology used detracted from providing a clear understanding of the nature of the signal. I felt it would have been better to stress that we are dealing with a scattering signal that is quite separate from the normal usage of the satellite's measurements.

If one wanted to generate a quantitative analysis of the signals, I expect the most likely starting point would be SASKTRAN, the freely available analysis software from the OSIRIS team that is geared toward a limb scattering geometry.

The nature of the aerosols is treated as a mystery (referred to as containing a strong liquid water content), but it really isn't. Volcanic eruptions are well known to create sulfate aerosols, liquid droplet mixtures of $H_2SO_4$ and $H_2O$. These aerosols contain a fraction of water, but the presence of sulfuric acid dissolved in the droplet changes the spectral response compared to pure water. The optical constants (real and imaginary components of the refractive index) for sulfate aerosols (aqueous H2SO4) are known. They have been measured in the laboratory. It is not appropriate to use optical constants for pure H2O as a gauge unless one expects liquid water droplets are generating the signal, which is not the case.

I have measured aerosols from the Tonga plume from the Atmospheric Chemistry Experiment. The figure below shows the measured (in blue) and fitted (in orange) results for aerosols observed a few weeks after the eruption [occultation ss99623, measured at latitude 16 S and longitude 166 W on February 7[th], 2022, at an altitude of 21.6 km]. The fitted results employ a set of sulfate aerosol optical constants [Lund Myhre CE, Christensen DH, Nicolaisen FM, and Nielsen CJ, Spectroscopic study of aqueous $H_2SO_4$ at different temperatures and compositions: variations in dissociation and optical properties. J Phys Chem A 2003;107:1979–1991, https://doi.org/10.1021/jp026576n]. The fact that the measurements can be reproduced accurately using optical constants for sulfate aerosols verifies the aerosol type as sulfate. Over the years that Tonga aerosols have persisted in the atmosphere, there has been some variation in the relative amount of H2SO4 and H2O in the droplets (driven by changes in temperature and ambient water vapor levels), but the aerosol type has remained unequivocally sulfate. You should use refractive index information for that aerosol type when evaluating the spectral response, not refractive index information from a different particle type (like pure liquid H2O droplets).

[Figure]

Above: observed and fitted results for ss99623 21.6 km, showing that the predominant stratospheric aerosol type following the Tonga eruption is sulfate.

As for ice, there is some evidence that the approach, using the selected wavelengths, might be somewhat blind to the presence of large ice particles.  Below, I have reproduced Figure 7 from the paper, showing the comparison of CALIOP observations with the ratio of signals at 1.61 and 2.25 microns.  Circled in red is a feature (around 16-17 km in the tropics) in the CALIOP observations that likely relates to high cirrus clouds, thin clouds composed of relatively large ice particles, a common occurrence in the tropical upper troposphere.  As mentioned in the paper, there does not appear to be corresponding signals for any of the tropical clouds in the 1.61/2.25 ratio.  While there is a time difference between the two data sets, which means the clouds might have dissipated, this could also point to a similar spectral response at the two wavelengths for ice particles, which would not generate a feature in the 1.61/2.25 ratio.

[Figure]

The figure below is reproduced from the Warren and Brandt paper describing optical constants for ice [Warren, S. G. and Brandt, R. E.: Optical constants of ice from the ultraviolet to the microwave: A revised compilation, Journal of Geophysical Research: Atmospheres, 113, 2008]. The arrow indicates the values for the two wavelengths employed in the ratio. They are quite similar, suggesting there could be a low contrast in the spectral response for ice at those wavelengths. A greater contrast could be achieved for ice by bringing in one of the other available wavelengths, such as 0.86 microns, but at the expense of dealing with larger scattering efficiencies, which might complicate the analysis.

[Figure]

In the reproduction of Figure 7 (2 figures up in this document), comparing CALIOP and the 1.61/2.25 ratio, the box added to the lower portion of the figure highlights a "waterfall" artifact in the results that arises from difficulties in separating out altitude information from the measurements (the reason the results look so diffuse relative to the sharply defined features from CALIOP). This will impact the altitude plots presented in Figure 10, yielding artificially inflated values for the ratio at lower altitudes. Although certainly beyond the scope of this paper, the geometry of the measurement could potentially lend itself to a tomographic analysis [Bourassa, A. E., Zawada, D. J., Rieger, L. A., Warnock, T. W., Toohey, M., & Degenstein, D. A. (2023). Tomographic retrievals of Hunga Tonga-Hunga Ha'apai volcanic aerosol. Geophysical Research Letters, 50, 2022GL101978 https://doi.org/10.1029/2022GL101978]. If different pixels on the imager provide measurements through the same plume at different angles, supplemented by different views through the plume as the Earth rotates below the geostationary satellite, you might be able to sharpen the altitude discrimination.

In summary, I think this dataset provides a great opportunity for a new aerosol product, if tools such as SASKTRAN were applied to analyze the limb scattering measurements. I felt the nature of the measurements (limb scattering rather than reflection) should have been emphasized more and promoted as the potential source of a new data product. I disagree with the approach of using the wrong optical constants for evaluating the spectral response, discussing 'high water content' in terms of refractive index information for pure H2O, when the aerosols are known to be sulfate, which has different (and known) refractive index information. For ice, I would suggest looking at the ratio of the signals at 0.86 and 2.25 microns to see if a feature appears in the plots from the presumed cirrus clouds. This would verify whether the 1.61/2.25 ratio might have a "blind spot" for ice.

---

## Author Response (AR2)

**Response to reviewer**

I am very grateful for this insightful and constructive review of my paper and the reviewer has raised some very interesting points which I address below. There are four main aspects of my paper that the reviewer questions. These are:

1. **Terminology.** The terminology used to describe the measurements. I concur with the reviewer that scattering is the correct terminology and as pointed out I used the term "reflectance" as this is commonly used in reference to shortwave radiation reflected from the surface, clouds and atmosphere and recorded by the Himawari-8 shortwave detectors. Consequently, I have altered the text from "reflectance" to "scattered light" in the appropriate places and added a footnote sentence to convey this aspect. The changes are highlighted in coloured text.

2. **Radiative Transfer.** It is correct to say that the paper would benefit from a detailed radiative transfer analysis and this was stated in the paper. I was not aware of the SASKTRAN code and I have thus added a reference to this code (Bourassa et al., 2008) as a means for performing a more detailed theoretical analysis. However, I maintain that this is beyond the scope of this "introductory" paper, which introduces a new and unintended use of Himawari measurements.

3. **Composition.** The reviewer suggests that somehow I think the composition of the stratospheric aerosol is a "mystery". It was not my intention to address the question of the exact composition of the aerosol. From a scientific methodology standpoint, in my view, it is not a good approach to assume one knows the result before performing the analysis. Sure, there is evidence that the aerosol is predominantly sulfate but why should I assume that for my new data analysis? A better approach (in my view) is to assume one does not know a priori the composition of the aerosol and then provide evidence one way or the other. In any case, I think the reviewer has misunderstood the point of the 1.61/2.25 ratio. This was intended to discriminate ice and water, since it is possible that some of the scattered light analysed could be from ice cloud rather than from the liquid water content of the stratospheric aerosol. High level ice cloud can penetrate the tropopause and so scattered light from these clouds may be misinterpreted as being from the volcanic aerosol. I don't think using refractive index data for sulfate provides any more information – as the plot below (Fig.1R) shows, but it does suggest that a sulfate aerosol would give a similarly high value for the ratio.

[Figure]

Fig. 1R. Ratio of the imaginary parts of the refractive index of Ice/Water and Ice/sulfate

Fig. 1R is the same as the m/s Fig. 4 but now including a ratio using refractive index data (from the Oxford U. ARIA data base, which uses the Palmer and Williams, 1975 measurements) for sulfate and with an expanded abscissa. The same (qualitative) conclusion can be made about the ice content of the aerosol based on the 1.61/2.25 µm ratio for sulfate rather than pure water. There may, admittedly, be quantitative differences, but as noted this could only be properly assessed using a radiative transfer approach. I have added a new sentence to convey this finding.

Not all volcanic eruptions create sulfate aerosols, the eruption of Chaiten in Chile is a good example (Prata et al., 2015). The material, which reached as far as Australia was predominantly particulate fine ash, probably andesitic in composition. Likewise, the Australian bushfires of 2019/2020 created a stratospheric aerosol (Khaykin et al., 2020) that contained particulates and likely water/ice and oxides of C and N. As part of this research, I analysed copious amounts of data, not just for Hunga but for some other eruptions and for other aerosol types (smoke form fires). I also investigated different channel combinations and indeed developed some interesting metrics by combining 3 channels. These analyses and others were not included in this paper for the sake of brevity; I was predominantly concerned that the ratio might ve seeing some other aerosol or ice clouds that had penetrated the tropopause. To show that other aerosols are indeed detected, I include a new figure here for the 2019/2020 Australian bushfires using a different channel ratio – clearly much more work could be done with these data.

[Figure]

Fig. 2R. Detection of aerosol layer due to bushfire smoke on 13 January 2020. Note that a different channel ratio 1.61/0.64 µm is used rather than 1.61/2.25 µm.

4. **Ice detection and vertical resolution**. I think the reviewer is essentially stating (perhaps more elegantly) what I intended that the ratio is less sensitive to ice and so is detecting the liquid water content of the aerosol. In my analyses low values of the ratio are either because there is nothing scattering the light back or it is ice doing the scattering, since the ratio of imaginary parts of the refractive indices (related to absorption or 1-scattering, in simplified terms) is a measure of the liquid water content. I didn't suggest that the ratio is sensitive to sulfate; although it likely is. The ratio is only semi-quantitative; radiative transfer calculations using limb geometry might provide a lower limit for the ratio, below which the composition could be inferred to be ice rather than liquid water (or sulfate). I made a plot of the ratio of 2.25/0.86 µm for 7 Feb 2022 and this is shown below.

[Figure]

Figure 3R.  Ratio of 2.25/0.86 µm reflectances for 7 Feb. 2022

Some features have changed – the suspect ice cloud between 10-14 km is clearly discernible. This presumably corresponds to the Caliop feature between 16-17 km. But it's not clear to me how this helps?  According to this ratio a high contrast is also expected for water clouds.  The idea of the ratioing was to find a measure that enhanced the water signature while diminishing the ice signature (it does not seem possible to discriminate a sulfate signature without resorting to the use of the thermal channels and here it is difficult because the measurement would be one of emission against a cold background and consequently poor SNR).  Nevertheless, I am not arguing that this is an optimal ratio or even the most appropriate, only that it seems to be able to detect the aerosol quite well.

The vertical resolution of the limb measurements is poor and I accept the reviewer's contention that some of the signal is an artefact of the analysis.  It's certainly an interesting idea to try to sharpen the measurements, but I'm not sure the methodology suggested would work since the Earth and satellite rotate at the same rate.  Some kind of 'dithering' may be possible using the intrinsic wobble of the satellite platform together with the high temporal sampling (10 minutes).  I believe satellite attitude data are available.

Again, I thank the reviewer for his very interesting comments and I suspect he knows that more research is required (by me, or hopefully someone else!) in order to give this new use of the Himawari data more 'legs'.

References

Bourassa, A. E., Degenstein, D. A., & Llewellyn, E. J. (2008). SASKTRAN: A spherical geometry radiative transfer code for efficient estimation of limb scattered sunlight. *Journal of Quantitative Spectroscopy and Radiative Transfer*, *109*(1), 52-73.

Khaykin, S., Legras, B., Bucci, S. *et al.* (2020) The 2019/20 Australian wildfires generated a persistent smoke-charged vortex rising up to 35 km altitude. *Commun Earth Environ* **1**, 22. https://doi.org/10.1038/s43247-020-00022-5

Palmer, K. F. and Williams, D., (1975) "Optical Constants of Sulfuric Acid; Application to the Clouds of Venus?," Appl. Opt. 14, 208-219.

Prata, A. T., S. T. Siems, and M. J. Manton (2015), Quantification of volcanic cloud-top heights and thicknesses using A-train observations for the 2008 Chaitén eruption, J. Geophys. Res. Atmos., 120, 2928–2950, doi:10.1002/2014JD022399.

---

## Author Response (AR3)

Hi Christian

Thanks for your suggestions.  I have modified the m/s according to them and it has removed some errors and consequently improved the paper.

Here are the changes:

1.  Line 9:  I've added the time period "Between January and April 2022 ..."
2.  Line 99: Fixed.
3.  Fig.3: I've removed te "Top", "Middle" and "Bottom" as this seems redundant now that I have (a), (b) and (c).
4.  Line 168:  Quite right.  I've removed the reference.
5.  Line 184:  Yes I agree this does sound contradictory.  Instead, I've just said the aerosol is elevated between 15-25 S.

Again thanks for guiding my paper through the peer review process – I know it is time consuming and takes you away from your research activities and it is much appreciated by me.

Kind regards

Fred

6 May 2024